# Fatty acid synthesis suppresses dietary polyunsaturated fatty acid use

Anna Worthmann[1], Julius Ridder[2], Sharlaine Y. L. Piel[2], Ioannis Evangelakos ®[2], Melina Musfeldt[2], Hannah Voß[3], Marie O'Farrell[4], Alexander W. Fischer ®[1,5,6], Sangeeta Adak[7], Monica Sundd[8], Hasibullah Siffeti[2], Friederike Haumann[2], Katja Kloth[2], Tatjana Bierhals[2], Markus Heine[1], Paul Pertzborn[1], Mira Pauly[1], Julia-Josefine Scholz[2], Suman Kundu ®[9], Marceline M. Fuh[1], Axel Neu[10], Klaus Tödter[1], Maja Hempel ®[2,11], Uwe Knippschild ®[12], Clay F. Semenkovich ®[7], Hartmut Schlüter ®[3], Joerg Heeren ®[1], Ludger Scheja[1], Christian Kubisch[2] & Christian Schlein ®[2] ✉

Dietary polyunsaturated fatty acids (PUFA) are increasingly recognized for their health benefits, whereas a high production of endogenous fatty acids – a process called de novo lipogenesis (DNL) - is closely linked to metabolic diseases. Determinants of PUFA incorporation into complex lipids are insufficiently understood and may influence the onset and progression of metabolic diseases. Here we show that fatty acid synthase (FASN), the key enzyme of DNL, critically determines the use of dietary PUFA in mice and humans. Moreover, the combination of FASN inhibition and PUFA-supplementation decreases liver triacylglycerols (TAG) in mice fed with high-fat diet. Mechanistically, FASN inhibition causes higher PUFA uptake via the lysophosphatidylcholine transporter MFSD2A, and a diacylglycerol O-acyltransferase 2 (DGAT2)-dependent incorporation of PUFA into TAG. Overall, the outcome of PUFA supplementation may depend on the degree of endogenous DNL and combining PUFA supplementation and FASN inhibition might be a promising approach to target metabolic disease.

The association of aberrant fatty acid synthesis and metabolic diseases has been investigated intensively in preclinical and clinical studies[1,2]. Obesity, insulin resistance, type 2 diabetes mellitus (T2DM), hyperthyroidism, menopause and pharmaceuticals can cause increased DNL in the liver[3–6]. Elevated DNL is closely linked to fatty liver, which is highly prevalent in the general population and frequently progresses to NASH[7]. So far, no medication has been approved for NASH, but recent studies examined pharmacologic inhibition of DNL as a treatment for targeting fatty liver diseases[8–11].

[1]Department of Biochemistry and Molecular Cell Biology, University Medical Center Hamburg-Eppendorf, Hamburg, Germany. [2]Institute of Human Genetics, University Medical Center Hamburg-Eppendorf, Hamburg, Germany. [3]Section / Core Facility Mass Spectrometry and Proteomics, University Medical Center Hamburg-Eppendorf, Hamburg, Germany. [4]Sagimet Biosciences Inc., 155 Bovet Rd., San Mateo, CA 94402, USA. [5]Department of Molecular Metabolism, Harvard T. H. Chan School of Public Health, Harvard University, Boston, MA, USA. [6]Department of Cell Biology, Harvard Medical School, Boston, MA, USA. [7]Division of Endocrinology, Metabolism & Lipid Research, Department of Medicine, Washington University, St. Louis, MO, USA. [8]National Institute of Immunology, New Delhi, India. [9]Department of Biochemistry, University of Delhi South Campus, New Delhi 110021 and Department of Biological Sciences, Birla Institute of Technology and Science Pilani, K K Birla Goa Campus, Goa 403726, India. [10]Department of Pediatrics, University Medical Center Hamburg-Eppendorf, Hamburg, Germany. [11]Institute of Human Genetics, University Hospital Heidelberg, Im Neuenheimer Feld 440, 69120 Heidelberg, Germany. [12]Department of General and Visceral Surgery, University Hospital Ulm, Ulm, Germany. ✉e-mail: c.schlein@uke.de

Fatty acid synthesis is performed by the multi-enzymatic protein FASN, producing the saturated fatty acid (SAFA) palmitate, a 16-carbon fatty acid without double bonds (16:0). Excess levels of palmitate are implicated in cellular stress and thus, the rate of palmitate production in the liver is thought as one possible mechanism to link DNL to metabolic disease[12]. In contrast, the rate of fatty acid synthesis in adipose tissue is associated with metabolic health including high insulin sensitivity and glucose tolerance[13]. A considerable amount of endogenously synthesized SAFA are desaturated by steaoryl-CoA-desaturase (SCD) to produce the potentially beneficial mono-unsaturated fatty acid (MUFA) palmitoleate (16:1)[14,15]. Notably, genetically engineered mouse models revealed that homozygous, and even heterozygous null-mutations of *Fasn* are embryonically lethal[16], identifying *Fasn* as a loss-of-function intolerant gene, which is also valid for the human situation as shown by a pLI score of 1.0 in exome/genome sequencing of more than 100,000 individuals in the gnomAd project[17]. To circumvent this problem, different tissue-specific *Fasn* knockouts and mice lacking lipogenic transcription factors such as carbohydrate response element binding protein (ChREBP)[1] have been established to study the physiological role of *Fasn* and DNL.

In humans, the impact of DNL on lipid classes has been mostly studied by stable isotope incorporation studies or proton magnetic resonance spectroscopy[18-20] and only rarely in tissue biopsies[21]. Moreover, human DNL has been proposed to account for up to 26% of liver triglycerides[22], whereas it accounts for 50% of lipids in mice[23] and even up to 80% in pigs[24].

In contrast to endogenously synthesized SAFA, which can cause lipotoxicity in the liver[14,25], it has been proposed that dietary PUFA can directly inhibit lipogenesis[26,27] and thus reduce plasma triglycerides[28,29], cholesterol[30] and liver steatosis[31]. These observations made PUFA supplementation a viable option for the treatment of metabolic disease. There are just very few determinants of PUFA storage known for the liver. One of the best understood genetic factors are variants in *PNPLA3*. PNPLA3[Ile148Met] carriers show increased liver lipid storage with increased PUFA to SAFA ratios[32] by reduced transfer of PUFA from triacylglycerols (TAG) to phospholipids[33] and subsequent suppression of DNL[34]. However, due to the reduced lipid transfer activity, this variant predisposes for the risk of cirrhosis and—despite higher amounts of PUFA – has no associated beneficial metabolic outcome.

To date, no large double-blinded, multicentric studies have been performed testing the effect of omega-6 or omega-3 PUFA on the disease progression of NASH. However, small trials with low dose omega-3 PUFA treatment or comparison of low-dose omega-3 to omega-6 dosing in patients with NASH were carried out and gave inconsistent results[35-39]. In a large primary prevention study low dose omega-3 PUFA failed to lower the risk for major cardiovascular events, although subgroup analyses suggested a benefit for African Americans and those with low fish consumption[40]. Furthermore, the REDUCE-IT study showed a lower risk for ischemic events and cardiovascular death after combined statin and omega-3 PUFA treatment in hypertriglyceridemic patients[41], whereas the STRENGTH study reported different results[42]. Of note, the reasons for different responses to PUFA treatment in distinct patient subgroups are largely unknown.

Here, we present evidence in vivo and in vitro that low endogenous DNL leads to high incorporation of PUFA. Lipidomic plasma analysis of a 5-year-old patient with a hypofunctional de novo *FASN* variant revealed a reduction in SAFA and MUFA in favor of a substantial increase of highly desaturated PUFA in plasma lipids. Moreover, we validated the relationship of DNL and PUFA abundance in TAG in a clinical trial using a human FASN inhibitor, as well as in mouse and cell culture models with genetically or pharmacologically induced low DNL. In summary, our data indicate that endogenous DNL controls the use of dietary PUFA in the liver. These results may have direct clinical implications for the selection of patients to be treated with PUFA and opens new avenues for studying the pathophysiology of DNL-associated liver damage.

## Results

### Genetic analysis of a *FASN*[Arg2177Cys] variant

The index patient, a 5-year-old boy, first of two children of non-consanguineous healthy parents from Kongo and Angola, presented with moderate delay in speech development, autistic behavior and macrocephaly. A trio whole-exome analysis revealed a putative pathogenic de novo missense variant in *FASN* (c.6529C>T, p.Arg2177-Cys) (Fig. S1a – for detailed medical history and genetic predictions see supplementary information). The missense variant was located at a conserved amino acid position of the acyl carrier protein (ACP) domain that binds the newly synthesized fatty acid within the multi-enzymatic complex FASN (Fig. S1b).

Patient index fibroblasts exhibited reduced FASN activity (Fig. 1a), as assessed by a reduction of [14]C-acetate incorporation into TAG, suggesting a functional impairment of the variant. However, as indicated by unchanged malonyl-CoA levels, no substantial concomitant substrate accumulation was observed (Fig. S1c), albeit malonylation of proteins was slightly enhanced (Fig. S1d) and no effects on *FASN* gene expression were detected (Fig. S1e). Protein turnover studies revealed that the FASN[Arg2177Cys] variant overexpressed in HEK cells showed a decreased protein stability (Fig. 1b, c), which has been described to be mediated by increased ubiquitinylation and proteasomal degradation after FASN hyperacetylation[43]. Indeed, the arginine to cysteine substitution caused an increased ubiquitinylation of HA-tagged FASN (Fig. 1d, mass spec confirmed ubiquitinylation sites in Fig. 1e), as well as increased acetylation (Fig. 1f—mass spectrometry confirmed increased acetyl-lysine sites and quantitative differences are indicated in Fig. S1b and Fig. S1f). Importantly, the acetylation could be further increased in vitro by incubating immunoprecipitated FASN with acetyl-CoA at 37 °C (Fig. 1g), which reflects a mechanism of non-enzymatic self-acetylation of FASN. Non-enzymatic acetylation was recently described as a novel mechanism of protein acetylation in mitochondrial membranes but has not been described as a variant pathomechanism[44]. Interestingly, when co-expressing FASN[Arg2177Cys] and wild-type FASN, the wild-type FASN showed a higher protein turnover as well (Fig. S1g, h). This might be explained by proteasomal degradation of the full FASN heterodimer, because FASN[Arg2177Cys] did not transacetylate the wild-type protein (Fig. S1i). Of note, overexpression of the FASN deacetylase *HDAC3*[43] resulted in reduced FASN[Arg2177Cys] acetylation (Fig. 1h) and subsequently, reduced ubiquitinylation (Fig. 1i) and stabilized degradation (Fig. S.1j). Together, the functional analysis of the mutant revealed that the missense variant leads to hyperubiquitinylation, which results in hypofunctionality and compromises protein stability (models in Fig. S1k, l).

### Lipidomics of genetic and pharmacological FASN modulation

To study functional consequences of the mutation in vivo, we used a lipidomic approach for analyzing plasma samples of controls and the index patient. Firstly, we estimated the residual function of the *FASN* variant in vivo by calculating the lipogenic index, a ratio of a mainly DNL-derived fatty acid (palmitoleate, 16:1) to a strictly dietary fatty acid (linoleic acid, 18:2) (Fig. 2a, Fig. S2a–d) and the ratio of essential to non-essential fatty acids (Fig. S2e). Compared to healthy controls (adult and child controls) and the patient's healthy father, the lipogenic index was reduced by half in the patient carrying the *FASN* mutation (Fig. 2a, Fig. S2a–d, f), suggesting a robust reduction of the FASN activity also in vivo. The essential (dietary) to non-essential fatty acid ratio was slightly higher in the patient (Fig. S2e). Plotting the relative changes in all plasma lipid species of the patient compared to controls, we found that smaller (14C-16C) and low saturated fatty acids (0-1 double bonds) were decreased, while especially highly unsaturated fatty acids, such as docosahexaenoic acid (DHA, 22:6), were strongly increased in the top 30 upregulated/downregulated species investigated (Fig. S2g). As the fatty acid composition of plasma TAG of fasting individuals correlates with liver TAG fatty acids, we calculated the total amount of each, saturated, monounsaturated and polyunsaturated fatty acid-containing TAG

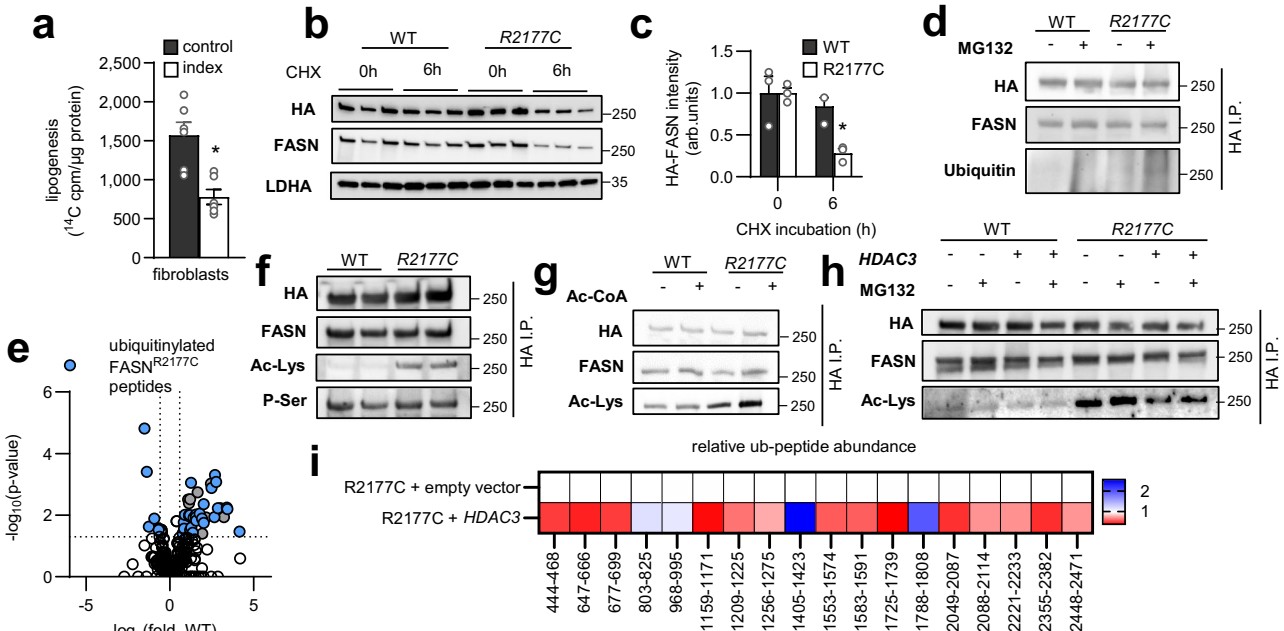

**Fig. 1 | Functional analysis of a FASN^Arg2177Cys variant. a** FASN activity in patient-derived fibroblasts (index) and age-and sex matched controls (control) estimated by incorporation of $^{14}$C-acetate into lipid fraction of fibroblasts (data are presented as mean ± SEM, $n = 6$, unpaired two-tailed Student's $t$ test *$P = 0.0022$). **b** Western Blot against HA, FASN and LDHA of cycloheximide (CHX) incubated HEK293T-cells expressing HA-tagged wild type *FASN* (*WT*) or HA-tagged patient variant *R2177C-FASN* (*R2177C*) and (**c**) western blot quantification (data are presented as mean ± SEM, $n = 3$, unpaired two-tailed Student's $t$ test *$P = 0.0094$). **d** Western blot against HA, FASN and ubiquitin of HA-immunoprecipitated (HA I.P.) HA-tagged wild-type *FASN* (*WT*) or HA-tagged R2177C FASN (*R2177C*) from overexpressing HEK293T-cells treated with (+) or without (−) the proteasome inhibitor MG132. (**e**) Volcano plot showing significantly differentially abundant ubiquitinylated peptides between immunoprecipitated FASN from WT (WT) and R2177C-FASN (FASN^R2177) cells after incubation with MG132 ($n = 4$; two-way ANOVA highlighted are peptides with corresponding $P$-value < 0.05 and fold change difference > 1.5). **f** Western blot against HA, FASN, acetyl-lysine (Ac-Lys) and phospho-serine (P-Ser) of HA-tag immuno-precipitation (HA I.P.) in HEK293T cells overexpressing HA-tagged wild-type *FASN*

(*WT*) or HA-tagged *R2177C-FASN* (*R2117C*) constructs. Western blots are representative picture of three independent experiments with the same result. **g** HA-tagged wild-type *FASN* (*WT*) or *R2177C-FASN* (*R2177C*) was overexpressed in HEK293T cells and after HA-immunoprecipitation (HA I.P.), WT and variant FASN was incubated with (+) or without (−) acetyl-CoA (Ac-CoA) in vitro for 6 h at 37 °C and western blot against HA, FASN, and acetyl-lysine (Ac-Lys) was performed. Western blots are representative picture of three independent experiments with the same result. **h** Western blots against HA, FASN and acetyl-lysine (Ac-Lys) of HA-tag immunoprecipitation (HA I.P.) in HEK293T cells expressing HA-tagged wild-type *FASN* (*WT*) or R2177C variant *FASN* (*R2177*) and *HDAC3*, a FASN deacetylase, co-treated with (+) or without (−) the proteasome inhibitor MG132. Western blots are representative picture of three independent experiments with the same result. **i** Ubiquitinylated FASN peptide abundance of immunoprecipitated R2177C-FASN overexpressed in HEK293T-cells in combination with a control (R2177C + empty vector) or *HDAC3*-vector (R2177C + *HDAC3*) ($n = 4$, data are shown as fold change vs. R2177C + empty vector). Source data are provided as a Source Data file.

species. Compared to controls, the omega-6 and omega-3 PUFA-containing TAG were higher in plasma of the index patient, in both composition and in part also in concentration (Fig. 2b, S3a–c). Interestingly, some elongation ratios, a marker of alternative malonyl-CoA use, were slightly higher in plasma and fibroblasts of the patient (Fig. S3d–e).

To investigate a causal relationship between FASN activity and PUFA abundance in TAG in humans, we performed lipidomic analysis of plasma samples from fasted individuals with NASH derived from the FASCINATE-1 trial[10]. Of note, in these samples, plasma TAG are mainly derived from liver-secreted VLDL, which are subsequently highly influenced by adipose tissue lipolysis. Importantly, individuals with NASH receiving 50 mg of the oral FASN inhibitor TVB-2640 for 12 weeks showed lower steady-state SAFA, whereas PUFAs in TAG composition were substantially higher compared to patients receiving placebo (Fig. 2c). Of note, these PUFA differences occurred in the omega-6 PUFA linoleic acid (18:2), but also the omega-3 PUFA 20:5 (EPA), and by trend in 22:6 (DHA) (Fig. 2d).

### Fatty acid preference in triacylglycerols is controlled by FASN, DGAT2 and MFSD2A

To gain mechanistic insights into TAG composition biology during genetic or pharmacologic FASN inhibition, we identified possible pathways affected by FASN inhibition, which are PUFA uptake, oxidation or retention. As DHA was robustly affected in the index patient,

and also after pharmacological FASN inhibition, and since DHA is widely used in supplementation trials in humans, we focused on DHA biology. Interestingly, HuH7 cells treated with the FASN inhibitor TVB-2640 showed increased DHA uptake into the cells (Fig. 3a). While oleate showed reduced oxidation upon FASN inhibition, DHA oxidation, which—in contrast to oleate—is mainly degraded in peroxisomes, was not affected (Fig. 3b). Of note, inhibiting FASN during in vitro lipoprotein secretion did not change the amounts of oleate and DHA released into the supernatant (Fig. S4a–c). However, when FASN was inhibited during lipid loading, higher amounts of DHA per triglyceride were detected in the supernatant after VLDL secretion (Fig. 3c; Fig. S4d, e), suggesting that higher DHA uptake (Fig. 3a) was leading also to higher secretion. Stable isotope tracing experiments revealed that upon FASN inhibition, HuH7 cells treated with equimolar doses of deuterated palmitate, oleate and DHA showed lower palmitate incorporation into secreted TAG species whereas oleate incorporation was unaltered. Of note, DHA incorporation was higher than oleate or palmitate (Fig. 3d). After addition of etomoxir to control for possible beta-oxidation effects, these effects were persistent in secreted TAG (Fig. 3d), whereas in cell pellets, additional etomoxir treatment resulted in a higher incorporation of deuterated DHA compared to oleate or palmitate (Fig. S4f). To identify the enzymes promoting the incorporation into TAG, we inhibited FASN with TVB-2640 and performed a knockdown screening of enzymes involved in TAG synthesis.

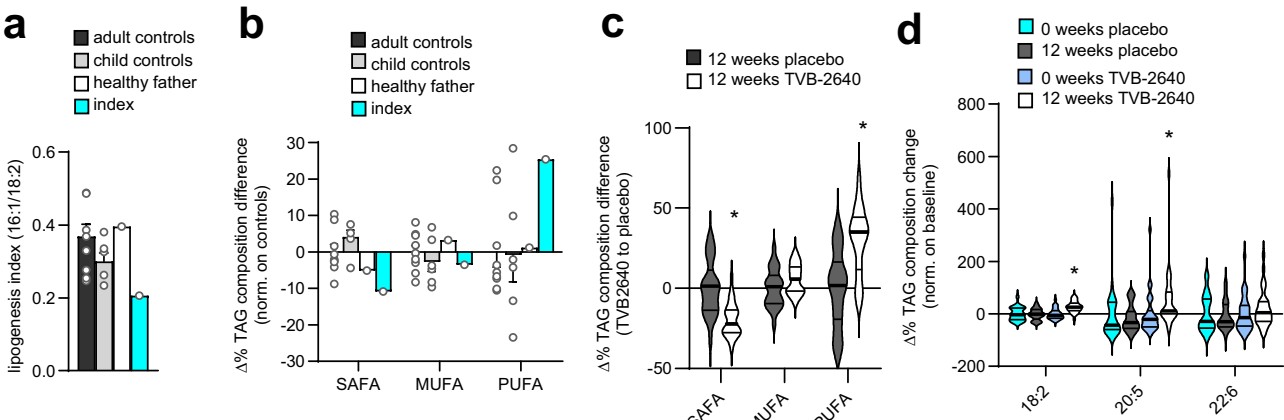

**Fig. 2 | Lipidomics of genetic and pharmacological FASN modulation. a** Plot of the DNL index, a marker to estimate the rate of DNL which is given as the ratio of (endogenously producible) palmitoleate (16:1) to strictly dietary linoleic acid (18:2) in plasma TAG in the index patient (index), his healthy father, a non-carrier, (healthy father) and adult (adult controls) and child controls (child controls) (data are presented as mean ± SEM for adult control (*n* = 9) and child controls (*n* = 6). **b** Differences in plasma TAG composition per each fatty acid class of the index patient (index) and different controls (adult controls, child controls, healthy father) (data are presented as mean ± SEM for adult control (*n* = 9) and child controls (*n* = 6). **c** Differences in plasma TAG composition per fatty acid class in NASH

patients treated with 50 mg of the FASN inhibitor TVB-2640 compared to NASH patients receiving placebo for 12 weeks (data are shown as violin plot, *n* = 25 for placebo and *n* = 28 for TVB-2640, unpaired two-tailed Student's *t* test, SAFA: *\*P* = < 0.0001; PUFA: * *P* = < 0.0001. **d** Changes in omega-6 (18:2) and omega-3 (20:5; 22:6) fatty acid residues in the TAG fraction of plasma samples from NASH patients treated with placebo (12 weeks placebo) or 50 mg of the FASN inhibitor TVB-2640 (12 weeks TVB-2640) for 12 weeks compared to baseline (0 weeks placebo and 0 weeks TVB-2640) (data are shown as violin plot, *n* = 25 for placebo and *n* = 28 for TVB-2640, paired two-tailed Student's *t* test, 18:2: * *P* < 0.0001; 20:5 *\*P* = 0.0350. Source data are provided as a Source Data file.

Importantly, FASN inhibition led to higher PUFA composition in TAG (Fig. 3e). Interestingly, a 24 h knockdown of *DGAT2*, but not *AGPAT1*, *AGPAT2, DGAT1, LPIN1* or *GPAM* was sufficient to reduce the higher PUFA composition induced by FASN inhibition (Fig. 3e). Of note, the decreased capacity to store PUFAs in TAG after *DGAT2* knockdown lead to increased secretion of DHA into the supernatant (Fig. 3f–g). Moreover, DHA uptake and secretion were diminished by the knockdown of the gene encoding the lysophosphatidylcholine transporter major facilitator superfamily domain-containing protein (MFSD2A) (Fig. 3f–g), which also resulted in slightly higher DNL gene expression (Fig. S4g) and a slightly higher 16:1 TAG composition in *MFSD2A* knockdown cells (Fig. 3h). Importantly, the PUFA-raising effect of TVB-2640 on TAG composition was abrogated in *MFSD2A* knockdown cells, suggesting that increased PUFA uptake (indicated in Fig. 3a) is an important mechanism for the proposed higher PUFA use in states of low FASN activity. The MFSD2A-dependent diminished effect of FASN inhibition on PUFA was also seen on the concentration level of highly desaturated fatty acids in TAG (Fig. 3i), giving further evidence for an active process increasing the PUFA use after FASN inhibition (Fig. 3a, h, i). Despite reduced oleate oxidation, FASN inhibition resulted in a reduced TAG concentration (Fig. S4h), indicating that oxidation of fatty acids accounts only for a minor role in the amount of lipid storage in this setting.

To further understand the role of DGAT2 in the context of non-lipoprotein secreting cells, we incubated mouse embryonic fibroblasts (MEFs) of wild type and *Dgat2⁻/⁻* mice with equimolar levels of SAFA (16:0, 18:0), MUFA (16:1, 18:1) and PUFA (18:2, 22:6). This revealed a DGAT2-dependent preference for PUFA over MUFA and SAFA (Fig. S5a) with a clear selective preference for DHA (Fig. S5b). Importantly, 24 h of pharmacological FASN inhibition with C75, as an independent FASN inhibition approach, led to increased relative incorporation of exogenous 18:2 and 22:6 fatty acid residues in incubated TAG of wild type but not *Dgat2⁻/⁻* MEFs (Fig. S5c) supporting the hypothesis of a cell-autonomous phenotype. Fibroblasts from the patient carrying the variant of *FASN* also showed reduced TAG, a reduced DNL index and a higher TAG composition in PUFA (Fig. S6a–c). In contrast, overexpression of FASN in HEK293T cells

resulted in reduced PUFA levels in TAG concentration (Fig. S6d) and TAG composition (Fig. S6e) whereas SAFA and palmitoleate (16:1) were increased. This further emphasizes that high FASN activity actively suppresses PUFA abundance in cells.

To investigate whether other enzymes of the DNL pathway also show similar effects as FASN, we inhibited ATP-citrate lyase (ACLY), an enzyme producing acetyl-CoA as a precursor for acetyl-CoA carboxylase (ACC) in HuH7 cells using bempedoic acid. Albeit to a lower extend, DNL inhibition by bempedoic acid led to higher PUFA levels in TAGs (Fig. S7a).

### DHA incorporation is controlled by the DNL rate

To generalize our findings and to rule out effects seen by biological variation in humans, we also investigated preclinical in vivo models of genetic or pharmacologically low DNL. As heterozygous global *Fasn*-knockout mice have decreased viability, and the index patient exhibits a globally hypofunctional FASN, we used global *Chrebp⁻/⁻* mice, which show decreased DNL-associated transcription in several organs, such as liver (Fig. S8a), adipose tissue and gut[45]. Indeed, *Chrebp⁻/⁻* mice exhibited decreased MUFA and increased PUFA composition in plasma TAG (Fig. S8b, c) and liver fatty acid profile (Fig. S8d, e), confirming a direct link between reduced DNL and PUFA utilization. Plasma fatty acid levels of the only published missense (and putative gain-of-function) mouse model *Fasn*^R1812W[46] were not altered (Fig. S8f). To determine their use in plasma and liver, we injected *Chrebp⁻/⁻* mice intravenously with radioactively labeled fatty acids complexed to fatty acid-free BSA. While the incorporation of DHA into liver TAG and VLDL-particles was higher in *Chrebp⁻/⁻* mice compared to wild-type littermates, palmitate accumulated in liver TAG and did not show a flux towards VLDL in *Chrebp⁻/⁻* mice. Further, loss of *Chrebp* did not change oleate incorporation in compartments investigated and no oleate flux was detected into phospholipids (Fig. S8g–i).

*Chrebp⁻/⁻* mice have been described to show a reduced VLDL synthesis[47], which we confirmed by higher liver TAG and lower plasma TAG levels (Fig. S9a–d).

Intake of a high-fat diet (HFD) in preclinical models reflects a state of high SAFA, the main product of DNL. This raises the question if the

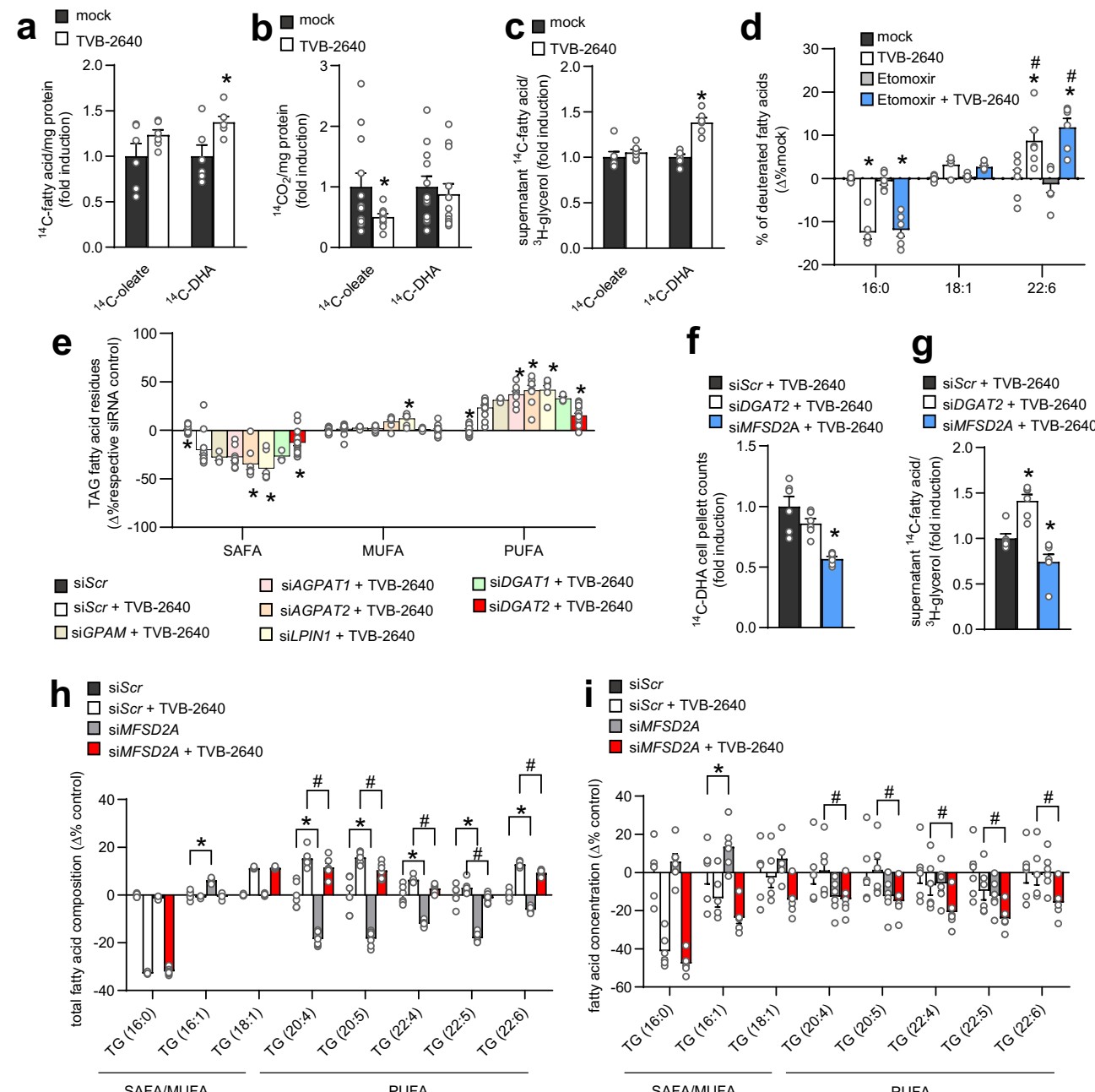

effects of PUFA incorporation described here are only relevant on low-fat diet (chow) or also important during HFD where many other dietary fatty acids compete with DHA for supplementation. As diet-induced obesity is associated with dyslipidemia, lipid-lowering PUFA therapy regimes are of clinical interest. To investigate whether DNL under conditions of exogenous high palmitate and oleate is still a major determinant of PUFA use, the following experiments were performed in mice fed a HFD. Indeed, wild-type mice on HFD showed no additional DHA-mediated plasma triglycerides lowering (Fig. S10a), but—likely due to higher fatty acid competition—exhibit reduced DHA incorporation (Fig. S10b) after DHA supplementation. Treatment with DHA was able to lower liver TAG on chow diet, but not on HFD (Fig. S10c). In contrast, compared to WT mice, HFD-fed *Chrebp*−/− mice showed strongly increased PUFA levels (Fig. 4a) and in particular the enhanced incorporation of supplemented DHA into plasma (Fig. S10d) and liver TAG (Fig. S10e). This was accompanied by an almost 50% reduction of plasma triglycerides, which did not occur in wild type mice (Fig. 4b), indicating an additional therapeutic DHA response

during HFD. The reduction in VLDL secretion was not further enhanced by DHA supplementation in *Chrebp*−/− mice (Fig. S10f, g) and glucose tolerance, insulin, body weight, FFA as well as hepatic gene expression analysis remained unaffected by the combination of *Chrebp*-knockout and DHA (Fig. S10h−n) indicating that for anti-inflammatory DHA effects in the liver, Chrebp activity is needed.

TAG of livers of *Chrebp*−/− mice showed decreased oleate and higher linoleate composition (Fig. S11a) on chow diet. When feeding a HFD, which is enriched in oleate and shows lower linoleate composition (Fig. S11b, c), differences in linoleate were blunted. When supplementing *Chrebp*−/− mice DHA during HFD, the DHA composition exceeded the wild-type littermate levels suggesting an actively regulated process (Fig. S11d, e). Interestingly, lipid composition of isolated hepatic ER of *Chrebp*−/− mice showed decreased SAFA and increased PUFA in LPE and TAG (Fig. S12a), whereas in PC, especially 18:2 species were increased (Fig. S12b). However, SREBP cleavage as a possible compensatory mechanism[48] was not different in the ER (Fig. S12c).

**Fig. 3 | Fatty acid preference in triacylglycerols is controlled by FASN, DGAT2 and MFSD2A. a** Fatty acid uptake assessed by scintillation counting of cell pellets from HuH7 cells pretreated with 500 nM of FASN inhibitor TVB-2640 (TVB-2640) or DMSO control (mock) for 24 h and incubated with radioactively labeled $^{14}$C-oleate or $^{14}$C-DHA for 4 h (data are shown as mean ± SEM; $n = 6$, unpaired two-tailed Student's $t$ test *$P = 0.0235$). **b** Fatty acid oxidation assessed by scintillation counting of $^{14}$C-$CO_2$ released by HuH7 cells treated with TVB-2640 (TVB-2640) or control (mock) and radioactively labeled $^{14}$C-oleate or $^{14}$C-DHA (data are shown as mean ± SEM; $n = 12$, unpaired Student's $t$ test *$P = 0.0440$). **c** To stimulate lipoprotein secretion, HuH7 cells were loaded with oleate or DHA mixed with radioactively labeled tracers overnight and were incubated with or without FASN inhibitor TVB-2640. After lipid loading, cells were extensively washed and incubated with TVB-2640 (TVB-2640) or control (mock). Supernatant was collected 5 h after inhibition and lipid extract was counted via scintillation counting (data are shown as mean ± SEM; $n = 6$, unpaired two-tailed Student's $t$ test *$P < 0.0001$). **d** HuH7 cells were incubated with equimolar amounts of deuterated palmitate (16:0), oleate (18:1) and DHA (22:6) and incorporation of deuterated fatty acids into TAG was measured in supernatant of the cells as a marker for secreted lipoproteins in control-treated cells (mock) or cells treated with a FASN inhibitor (TVB-2640), with a beta-oxidation inhibitor (Etomoxir) or the combination of both (Etomoxir+ TVB-2640) (data are shown as mean ± SEM $n = 6$; two-way ANOVA and Fisher's LSD test, (*) indicates significant differences versus respective mock, (#) indicates significant differences in 22:6 vs. respective group in 16:0 and 18:1 (16:0 TVB-2640: *$P < 0.0001$; 16:0 Etomoxir+ TVB-2640: *$P < 0.0001$; 22:6 TVB-2640 *$P < 0.0001$; 22:6 Etomoxir +TVB-2640: *$P < 0.0001$; 22:6 TVB-2640 vs. 16:0 TVB-2640: #$P < 0.0001$; 22:6 TVB-2640 vs. 18:1 TVB-2640: #$P = 0.0078$; 22:6 Etomoxir+ TVB-2640 vs. 16:0 Etomoxir +TVB-2640: #$P < 0.0001$; 22:6 Etomoxir + TVB-2640 vs. 18:1 Etomoxir +TVB-2640: #$P < 0.0001$. **e** TAG fatty acid class composition of HuH7 cells transfected with indicated siRNAs for 24 h and treated with TVB-2640 for another 24 h (data are shown as mean ± SEM; $n = 12$ for si$Scr$, $n = 12$ for si$Scr$ + TVB-2640, $n = 3$ for si$GPAM$ + TVB-2640, $n = 9$ for si$GPAT1$ + TVB-2640, $n = 9$ for si$GPAT2$ + TVB-2640, $n = 8$ for si$LPIN1$ + TVB-2640, $n = 3$ for si$DGAT1$ + TVB-2640, $n = 15$ for si$DGAT2$ + TVB-2640; two-way ANOVA and Fisher's LSD test (*) indicates significant differences

versus si$Scr$ + TVB-2640 (SAFA: si$Scr$ *$P < 0.0001$; si$AGPAT2$ + TVB-2640 *$P = 0.001$; si$LPIN1$ + TVB-2640 *$P < 0.0001$; si$DGAT2$ + TVB-2640 *$P = 0.0382$. MUFA: si$LPIN1$ + TVB-2640 *$P = 0.0136$. PUFA: si$Scr$ *$P < 0.0001$; si$AGPAT1$ + TVB-2640 *$P = 0.0021$; si$AGPAT2$ + TVB-2640 *$P < 0.0001$; si$LPIN1$ + TVB-2640 *$P < 0.0001$; si$DGAT2$ + TVB-2640 *$P = 0.0292$). **f** HuH7 cells were transfected with control siRNA (si$Scr$) or siRNA against $DGAT2$ (si$DGAT2$) or $MFSD2A$ (si$MFSD2A$), and treated with $^{14}$C-DHA as described in (**c**), and lipid extracts of cell pellets were counted via scintillation counting (data are shown as mean ± SEM; $n = 6$, one-way ANOVA Dunnett's multiple comparisons test, (*) indicates significant differences versus si$Scr$ + TVB-2640: si$MFSD2A$ + TVB-2640 *$P < 0.0001$. **g** HuH7 cells were transfected with control siRNA (si$Scr$) or siRNA against $DGAT2$ (si$DGAT2$) or $MFSD2A$ (si$MFSD2A$), and treated with $^{14}$C-DHA as described in (**c**), and lipid extracts of supernatants were counted via scintillation counting (data are shown as mean ± SEM; $n = 6$, one-way ANOVA with Dunnett's multiple comparisons test, (*) indicates significant differences versus si$Scr$ + TVB-2640: si$DGAT2$ + TVB-2640 *$P = 0.0014$; si$MFSD2A$ + TVB-2640 *$P = 0.0324$). **h** TAG fatty acid composition of HuH7 cells transfected with control (si$Scr$) or $MFSD2A$ siRNAs (si$MFSD2A$) for 24 h and treated without or with (+TVB-2640) for another 24 h (data are shown as mean ± SEM; $n = 6$, two-way ANOVA and Fisher's LSD test, (*) indicates significant differences versus si$Scr$ and (#) versus si$Scr$ + TVB-2640; 16:1: si$MFSD2A$ *$P < 0.0001$; 20:4: si$MFSD2A$ *$P < 0.0001$, si$MFSD2A$ + TVB-2640 #$P = 0.0159$; 20:5: si$MFSD2A$ *$P < 0.0001$, si$MFSD2A$ + TVB-2640 #$P = 0.0005$; 22:4: si$MFSD2A$ *$P < 0.0001$, si$MFSD2A$ + TVB-2640 #$P = 0.0185$; 22:5: si$MFSD2A$ *$P < 0.0001$, si$MFSD2A$ + TVB + 2640 #$P = 0.0037$; 22:6: si$MFSD2A$ *$P < 0.0001$, si$MFSD2A$ + TVB + 2640 #$P = 0.0223$). **i** Differences in TAG fatty acid concentrations of HuH7 cells transfected with control (si$Scr$) or $MFSD2A$ siRNAs (si$MFSD2A$) for 24 h and treated without or with TVB-2640 (+TVB-2640) for another 24 h (data are shown as mean ± SEM; $n = 6$, two-way ANOVA and Fisher's LSD test, (*) indicates significant differences versus si$Scr$ and (#) versus si$Scr$ + TVB-2640; 16:1: si$MFSD2A$ *$P = 0.0414$; 20:4: si$MFSD2A$ + TVB-2640 #$P = 0.0221$; 20:5: si$MFSD2A$ + TVB-2640 #$P = 0.0133$; 22:4: si$MFSD2A$ + TVB-2640 #$P = 0.0334$; 22:5: si$MFSD2A$ + TVB-2640 #$P = 0.0288$; 22:6: si$MFSD2A$ + TVB-2640 #$P = 0.0266$). Source data are provided as a Source Data file.

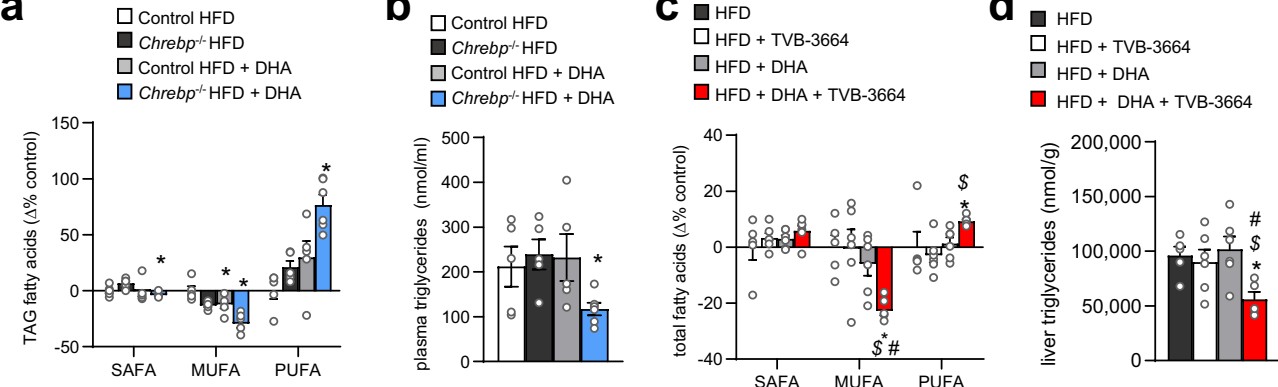

**Fig. 4 | Genetic or pharmacologic inhibition of DNL increases PUFA fatty acid composition in liver and positively affects DHA-supplementation therapy outcome. a** TAG fatty acid class composition of livers and (**b**) plasma TAG levels of 2 weeks HFD fed female $Chrepb^{-/-}$ mice ($Chrepb^{-/-}$ HFD, $n = 5$) or female wild type littermates (Control HFD, $n = 5$) without or with DHA supplementation for another week (Control HFD + DHA, $n = 5$; $Chrepb^{-/-}$ HFD + DHA, $n = 6$) (data are shown as mean ± SEM; two-way ANOVA and Fisher's LSD test, with (*) indicates significant differences versus respective non-DHA-supplemented group. **a** SAFA: $Chrepb^{-/-}$ HFD + DHA *$P = 0.0093$; MUFA: Control HFD + DHA *$P = 0.0241$, $Chrepb^{-/-}$ HFD + DHA *$P = 0.0031$; PUFA: $Chrepb^{-/-}$ HFD + DHA *$P = 0.0007$. **b** $Chrepb^{-/-}$ HFD + DHA

*$P = 0.0306$). **c** Liver fatty acid class profile and (**d**) liver TAG of male wild type mice fed a high-fat diet for two weeks and an additional week without or with DHA supplementation and receiving one week of daily TVB-2640 or control gavage (data are shown as mean ± SEM; $n = 5$ HFD, $n = 6$ HFD + TVB-3664, $n = 6$ HFD + DHA, $n = 6$ HFD + DHA + TVB-3664; significant differences were assessed using two-way ANOVA and Fisher's LSD test with (*) indicating significant differences vs. HFD, ($) showing significant differences vs. HFD + TVB-3664, and (#) showing significant differences vs. HFD + DHA. **c** MUFA: *$P = 0.0024$; $ $P = 0.0015$; # $P = 0.0141$; PUFA: *$P = 0.0459$; $ $P = 0.0087$. **d** *$P = 0.0159$ $ $P = 0.0293$ # $P = 0.0053$. Source data are provided as a Source Data file.

To investigate whether pharmacological inhibition of FASN has an impact on DHA supplementation, we fed wild type mice a HFD and treated them with TVB-3664, a mouse FASN inhibitor, and/or supplemented DHA to the food. Interestingly, on HFD neither DHA nor TVB-3664 alone was able to show an impact on total fatty acid composition

within the used short time frame of one week, but the double treatment increased PUFA in the fatty acid profile of the liver at the expense of MUFA (Fig. 4c, Fig. S13a). Importantly, TVB-3664 did not affect fatty acid profile in organs with generally lower FASN activity, such as muscle or heart (Fig. S13b–f). Adipose tissue however, showed a

slightly higher DHA composition on double treatment (Fig. S14a–b), albeit DHA storage and response to DHA supplementation was negligible compared to liver (Fig. S14c). Importantly, DHA treatment and double treatment caused a further reduction in DNL transcription in the liver and a lower presumably beneficial inflammatory gene expression pattern (Fig. S15a). Compared to the pro-inflammatory response of HFD + DHA− treated *Chrebp*−/− mice (Fig. S10m), this indicates that FASN inhibition reflects a superior pharmacologic target compared to transcriptional reduction of DNL. Interestingly, TVB-3664 monotreatment showed higher *Srebp1c* expression (Fig. S15a) and−in line with *Chrebp*−/− mice−reductions in *Mfsd2a*. Of note, malonyl-CoA levels did not change during FASN inhibition (Fig. S15b), whereas the malonylation of the proteome was significantly increased during FASN inhibition (Fig. S15c), which might buffer non-used malonyl-CoA accumulating due to inhibited DNL. Only double treatment of TVB-3664 and DHA was able to lower liver triglycerides, but not in monotherapies (Fig. 4d), whereas plasma triglyceride levels were reduced upon DHA treatment and were slightly lower in all treatment groups (Fig. S15d). HOMA-IR was reduced upon DHA and combined treatment compared to untreated mice without a significant impact on glucose level (Fig. S15e, f) indicating a slightly increased insulin sensitivity. Of note, on HFD DHA supplementation resulted in higher food intake, but lower body weight in mice (Fig. S15g, h).

In sum, by using different in vivo and in vitro approaches, we show that the incorporation of exogenous, dietary PUFA is controlled by endogenous DNL activity. The key finding, which is examined clinically and preclinically, is that DNL-derived fatty acids supersede dietary PUFA during TAG synthesis and thus control the PUFA incorporation. Whenever DNL is low, PUFA composition can increase and shows the highest preference of exogenous lipids (model in Fig. S16). The finding that DNL determines PUFA use critically impacts therapeutical interventions with PUFAs and their outcome and should be assessed in clinical studies and upon dietary and pharmacological PUFA supplementation.

## Discussion

In the present study, we describe a hypofunctional, pathogenic heterozygous *FASN* variant in a 5-year-old boy affected by the moderate delay in speech development, autistic-like behavior, and macrocephaly. We show that the FASN[Arg2177Cys] variant leads to protein instability and reduced activity of the fatty acid synthesis pathway. Whereas steady-state protein levels remained unchanged, the FASN variant showed greater ubiquitinylation, which might impact its enzymatic activity[49,50] given its increased protein turnover. Interestingly, the FASN variant showed a greatly enhanced acetylation close to the mobile ACP domain leading to the presumption of a non-enzymatic cysteine-induced acetylation recently described by James et al.[44] for mitochondrial membranes. Reduction of acetylation by HDAC3 was able to reduce the ubiquitinylation. However, as HDAC3 activity determines glucose uptake[51], beta-oxidation[51] and proteasome activity[52], HDAC3 overexpression is not a valuable target to rescue FASN protein function. Importantly, our study shows that cysteine missense variants can alter the protein stability also in the cytosol and that uncontrolled non-enzymatic acetylation per se can contribute to pathology. Of note, this might also be applicable to other pathogenic genetic variants, resulting in a cysteine amino acid substitution.

The hypofunctional genetic variant of FASN results in increased dietary PUFA composition preferably in TAG. The composition of TAG is used here to reflect the probability of a PUFA to be released upon hydrolysis, whereas high liver TAG concentrations e.g. in NASH[10] or due to genetic risk factors such as PNPLA3[Ile148Met 32–34] can be harmful, despite the fact they might show high PUFA concentrations. Interestingly, PNPLA3[Ile148Met] also leads to reduced DNL levels[34], which might additionally increase the PUFA abundance in the liver of these patients. At first view, the finding that FASN leads to high PUFA compositions

may seem somewhat paradoxical, as the missing endogenously produced palmitate, a SAFA and the initial product of FASN, is not replaced by a surplus of exogenous, dietary palmitate. However, this phenomenon was observed in the plasma from NASH patients treated with FASN inhibitor for 12 weeks, mouse models of both genetically or pharmacologically reduced DNL, and in cell culture models for mechanistical studies. Moreover, isotope tracing in HuH7 cells revealed that FASN inhibition resulted even in suppression of exogenous palmitate use, suggesting that not only endogenous, but also exogenous palmitate seems to enter the TAG compartment facilitated by FASN. We further show that upon low states of DNL, the molecular mechanism to increase the preference of exogenous PUFA in comparison to exogenous SAFA/MUFA in TAG seems to be orchestrated by MFSD2A-stimulated PUFA uptake and DGAT2-medited fatty acid esterification.

We demonstrate that FASN inhibition monotherapy of patients with NASH leads to increases in PUFA TAG composition after 12 weeks of oral administration highlighting that the described mechanism is applicable to disease states. Further, in HFD fed mice, we observed that the combination of DHA supplementation and FASN inhibition resulted in very rapid liver TAG lowering, whereas DHA supplementation in mice harboring a genetic deletion of the DNL transcription factor Chrebp lowered plasma TAG level. This discrepancy upon DHA supplementation between loss of *Chrebp* on the one hand and FASN inhibition on the other hand, might be explained by the fact that *Chrebp*−/− mice display an inherited reduced VLDL synthesis[47] which resulted in higher liver TAG and increased liver inflammation on HFD. Importantly, the substantial need for Chrebp activity for the anti-inflammatory effects of DHA rejects Chrebp as a potential pharmaceutical target in this setting but raises the question which Chrebp target is required to elaborate the anti-inflammatory potential of DHA.

In mice, FASN inhibition did not affect malonyl-CoA levels, but instead resulted in increased protein malonylation, which might serve as a malonyl-buffer within the cell or organ. Malonyl-CoA serves as a potent CPT1-inhibitor and may reduce fatty acid transport into the mitochondria to inhibit beta oxidation possibly causing lipid accumulation. Of note, despite similar malonyl-CoA levels upon FASN inhibition, we measured slightly decreased oxidation of mitochondrially oxidized oleate, but not peroxisomal oxidized DHA and in sum, no lipid accumulation. In fact, there was reduction of TAG in vitro and in vivo. This is of major importance for human therapy, as other DNL inhibitors lead to either hypertriglyceridemia or steatosis[8,9,53]. Our results in vitro and in vivo suggest that DNL inhibitors may increase responsiveness to PUFA therapy and emphasize the importance of assessing DNL in PUFA supplementation studies.

## Methods
### Ethical statement

All research presented in this study complies with the relevant ethical regulations. In particular, analysis involving material from study participants was carried out with written informed consent, and in the case of children with written, informed consent of their legally authorized representatives, with consent to publish identifiable information for the index patient and their parents and were conducted in accordance with the Declaration of Helsinki and approved by the ethical review board of the Ärztekammer Hamburg, Germany. Additionally, data was derived from the clinical study NCT03938246[10] was approved by the institutional review board of Advarra and conducted in accordance with ethical principles of the Declaration of Helsinki and consistent with the International Conference on Harmonization, Good Clinical Practice, and applicable regulatory requirements. Animal experiments performed in Hamburg were approved by the Animal Welfare Officers of University Medical Center Hamburg-Eppendorf (UKE) and the Behörde für Gesundheit und Verbraucherschutz Hamburg (N028/2020, date of approval: 2020/06/17;

N017/2021 date of approval: 2021/04/13) and animal studies at Washington University were approved by the Institutional Animal Studies Committee.

## Study participants

Exome sequencing on lymphocytes and further blood analyses were performed on the index patient and his healthy parents. In addition, fibroblast analysis of the index patient and two age- and sex-matched children controls with undiagnosed suspected genetic disease but absent metabolic phenotype and one adult healthy control volunteer was performed. All analyses were carried out with written informed consent and the study was approved by the local medical ethics committee. For lipidome analysis, plasma of both the father or nine sex- and age-matched controls of unknown ethnic origin have been used. FASCINATE-1 is a randomized 12-week placebo-controlled study of the FASN inhibitor TVB-2640 in NASH at 10 US sites (Clinical-Trials.gov number NCT03938246)[10]. Adults with ≥8% liver fat, and evidence of liver fibrosis by MR-elastography (MRE) ≥ 2·5kPa or liver biopsy were randomized to receive placebo or TVB-2640 orally, once-daily for 12 weeks. The primary endpoints were safety and relative change in liver fat, and lipidomics was an exploratory endpoint.

## Lipidomic analysis

Plasma, tissue and cell lipidomic analysis was performed using the Lipidyzer™ Platform from SCIEX. Briefly, samples were spiked with Lipidyzer™ Internal Standards (SCIEX) and lipid extraction was performed employing an adjusted MTBE/methanol extraction protocol[54]. Briefly, plasma, cell pellets and tissue samples were mixed with MTBE/methanol/water 10:3:2.5 (v/v/v). Following homogenization, samples were centrifuged (4 °C, 10,000 g, 10 min) and the upper phase containing the lipids was transferred into a new vial. Lipid extracts were concentrated and reconstituted in a mixture of dichloromethane (50):methanol (50) containing 10 mM ammonium actetate. Separation and targeted profiling of lipid species was performed combining differential mobility spectrometry and a QTRAP® system (QTRAP® 5500; SCIEX). Quantification of lipids was conducted by the Lipidyzer™ software (Lipidomics Workflow Manager software version 1.0.5.0; SCIEX) employing specific multiple reaction monitoring transitions. Tissues and cells were weight or protein normalized.

For the FASCINATE-1 study lipidomics of triacylglycerols, patient plasma was extracted with chloroform/methanol. The organic phase was dried, reconstituted in acetonitrile/isopropanol (1:1), centrifuged, and transferred to vials for UHPLC-MS analysis as previously described[55]. An appropriate test mixture of standard compounds was analyzed before and after the entire set of randomized sample injections in order to examine the retention time stability, mass accuracy, and sensitivity of the system. Metabolomics data were pre-processed using the TargetLynx application manager for MassLynx 4.1 (Waters Corp., Milford, MA).

## Stable isotope tracing

For stable isotope tracing, Huh7 cells were treated with equimolar levels (50 μM each) of d31-palmitic acid, d9-oleic acid, and d5-docosahexaenoic acid conjugated to 0.5 mM FA-free albumin in the presence of 500 nM TVB 2640, 100 μM Etomoxir, the combination of TVB-2640 and Etomoxir or control (DMSO) in DMEM supplemented with 10% lipoprotein-depleted serum and 1% penicillin-streptomycin. After 24 h, supernatants were collected on ice, cells were washed twice with ice-cold PBS containing BSA, and cells were harvested in PBS for the determination of protein concentration and lipidomic analysis. Lipids were extracted from supernatants and cells using an adjusted MTBE/methanol extraction[54] as described above. Extracted lipids were concentrated and reconstituted in a mixture of dichloromethane (50):methanol (50) containing 10 mM ammonium actetate. Shotgun lipidomic analysis was performed as described by Su et al.[56] with

additional added multiple reaction monitoring transitions to detect trace incorporation of deuterated fatty acids into TAG species of interest. In particular, the precursor mass of a TAG species expected to incorporate d31-palmitic acid was offset by 31 Da, while the product ions were offset depending on lipid class fragmentation patterns. In instances where double or triple incorporation was possible, the precursor mass was also offset by 62 or 93 Da respectively. Similarly, precursor masses were adjusted by 9 and 18 Da for d9-oleate and by 5 and 10 Da for d5-DHA.

## GC/FiD fatty acid profiling

Fatty acid composition in liver and plasma samples was determined by gas chromatography as described previously[21]. Fatty acids were extracted from liver samples according to the method of Folch. Briefly, 20 mg of liver tissue was mixed 400 μl of chloroform/methanol (2/1, v/v). After homogenization using a tissue lyser, the liver samples were centrifuged (1800 g, 5 min) and supernatants were transferred into a new vial. To prepare plasma fatty acid extracts, 50 μl of plasma were mixed with 100 μl internal standard mix (hepatdecanoic acid, tetradecanoate d27 and heptadecanoate d33, 200 μg/ml each in Methanol/Toluol 4/1) as well as 1000 μl Methanol/Toluol 4/1 and samples were vortexed vigorously. To prepare fatty acid methyl esters, 100 μl of the liver extracts were mixed with 100 μl internal standard mix (hepatdecanoic acid, tetradecanoate d27 and heptadecanoate d33, 200 μg/ml each in Methanol/Toluol 4/1) and both liver and plasma samples (total extracts) were mixed with 100 μL acetyl chloride. Samples were vortexed and heated in a capped tube for 1 h at 100 °C. After cooling to room temperature, 3 ml of 6% sodium carbonate was added. The mixture was centrifuged (1800 g, 5 min) and the upper layer was transferred to auto sampler vials. Gas chromatography analyses were performed using an HP 5890 gas chromatograph (Hewlett Packard) or a Trace 1310 gas chromatograph (Thermo Fisher) employed with the following stationary phase: DB-225 30 m × 0.25 mm i.d., film thickness 0.25 μm (Agilent) coupled to a flame ionization detector or a mass spectrometer (ISQ 7000 GC-MS, ThermoFisher Scientific, Dreieich, Germany) respectively. Peak identification and quantification were performed by comparing retention times and peak areas, respectively, to standard chromatograms and internal standards.

## Animal models

C57BL6/J WT mice were purchased from Janvier and *Mlxipl*-deficient mice on C57BL6/J background (*Chrebp*^−/− - mice) were purchased from Jackson Laboratories. Animal studies with these mice were carried out in Hamburg and were approved by the Animal Welfare Officers at University Medical Center Hamburg-Eppendorf and the Behörde für Gesundheit und Verbraucherschutz Hamburg (N028/2020, date of approval: 2020/06/17; N017/2021 date of approval: 2021/04/13). *Fasn*^R1812W mice on C57BL6/J background were generated as described before[46]. Studies with these mice were performed at Washington University and were approved by the institutional Animal Studies Committee. As data derived from human NASH patients was collected in male and female participants, animal experiments were performed in male and female mice (as indicated in the figure legends). Routinely, age matched (8–20 weeks) mice were kept in single cages in climate chambers (Memmert) with *ad libitum* access to regular chow diet (Altromin, 1329 P) and water. Mice were housed at room temperature (22 °C) with a day and night cycle of 12 h each and a humidity of 50%. FASN inhibition and DHA supplementation were performed after two weeks of high-fat diet (EF Bio-Serv F3282, Ssniff®, 60% fat) feeding via DHA supplemented high-fat diet (2.2%) at room temperature and daily oral gavage of 3 mg/kg TVB 3664 in 30% PEG400 for 7 days. Incorporation of radioactive fatty acids into VLDL was performed by body weight adapted i.v. injection of a solution containing 1 ml of 0.5 mM albumin conjugated with radioactive fatty acids (0.37MBq ^14C-DHA, 0.185 MBq ^3H-palmitate or 0.3 MBq ^3H-oleate per 10 mice) into

wild-type or *Chrebp*[-/-] mice, which additionally received a body weight adapted dose (5 μl/g) of tyloxapol (10% in NaCl) to inhibit lipases as described before[57]. Blood for EDTA plasma measurements was collected by cardiac puncture of anaesthetized mice.

## Gene expression

RNA isolation from tissues was performed by homogenizing tissues using TissueLyser (Qiagen) in peqGOLD TriFast (Peqlab). RNA was purified by NucleoSpin RNAII Kit (Macherey-Nagel) and cDNA was prepared using a High-Capacity cDNA Archive Kit (Applied Biosystems). Gene expression was assessed using Taqman assays supplied as assays-on-demands or SYBR green (Applied Biosystems), and data were normalized to the gene expression levels of the housekeeper *Tbp* for mice or *TAF1, RPLP0* or *GAPDH* for human samples as described before[21]. Validation of the sample and RNA and cDNA quality was assessed by 260/280 nM absorbance ratio measured with a Thermo Scientific NanoDrop™ and robust expression of housekeeping genes. Samples, which showed non-detectable housekeeping or target gene amplification were excluded. SYBR green assays were designed to avoid single nucleotide polymorphism sites and PCR products were designed to span exons, which preferably have long introns to reduce unspecific amplification of possible genomic DNA contamination. More information on Taqman assays as well as qPCR primer sequences can be found in the supplemental material.

## Cell culture

Cell culture studies were performed using fibroblasts derived from clinical patients, HEK293T cells purchased from ATCC (2017), HuH7 cells purchased from CLS (2022) and WT and *Dgat2*[-/-] MEFs kindly provided by the Walther/Farese lab (Harvard Medical School, Boston, USA). Fibroblasts of the index patient or age-matched and/or healthy controls were seeded into 6 cm dishes or 6-/12-well plates and were proliferated to confluence. Protein turnover studies were performed with 40-80 μmol Cycloheximide (Cat.-No. C4859, Sigma-Aldrich) for the indicated time frame. RNAi knockdown (information on used RNAi can be found in the supplemental material) and overexpression of wild type or FASN[Arg2177Cys] was performed via jetOPTIMUS® transfection of HEK293T cells. For overexpression studies, a FASN-pcDNA3.1 vector was purchased from Genscript or an *HDAC3*-pcDNA3.1 vector was purchased from Addgene. Immunoprecipitation studies were performed using Pierce™ Anti-HA Magnetic Beads, (Cat.-No. 88836, Thermo Scientific) incubated for 2 h or overnight at 4 °C. In vitro acetyl-CoA incubation was performed using HA-immunoprecipitated FASN incubated with 10 mM acetyl-CoA (Cayman Chemicals, Cat-no. 16160) as described in[44]. Wild type and *Dgat2*[-/-] mouse embryonic fibroblasts as well as HUH7 cells were cultured at 10% FCS, 4,5 g Glucose/l DMEM supplemented with 1% Penicillin/Streptomycin. Fatty acid incubations were performed by complexing the respective fatty acid (50 μM) to fatty-acid-free albumin in molar excess (0.5 mM) and supplementing the cells in lipoprotein-deficient FCS serum (LPDS) for 24 h. To pharmacologically inhibit FASN, cells were treated with 20 μM C75 FASN inhibitor (Cayman chemicals) or 500 nM TVB 2640 (Sagimet). InSolution™ MG132 was purchased from Calbiochem (Cat.-No. 474791) and was added for 6 h to the cell culture supernatant at a dose of 20 μM.

## DNL activity assays

[14]C-acetate was used to study DNL activity in fibroblasts. To this extend, fibroblasts were incubated with 10%FCS/4,5 g Glucose/l DMEM supplemented with 0.074 Bq [14]C-acetate. After 24 h, cells were washed with ice cold PBS, harvested in PBS and spun down at 100 g for 5 min. 1/5th of the pellet was used for BSA protein concentration measurement (Pierce™ BCA Protein Assay Kit). 4/5 of the pellet was used for lipid extraction using a modified Bligh and Dyer method described by Mc Donald et al.[58]. Briefly, the cell pellets were resuspended in 200 μl PBS

and mixed with 750 μl chloroform:methanol (1:2 v/v) followed by vigorous vortexing. After centrifugation (10 min, 10,000 g), the supernatants were transferred into fresh tubes and 250 μl chloroform and 250 μl PBS were added. After another step of vortexing, centrifugation (10 min, 10,000 g) induced a phase separation of two phases. The lower organic phase was taken and subsequently subjected to thin layer chromatography to isolate the triacylglycerol fraction. The triacylglycerol fraction was collected, and tracer amounts were counted by scintillation counting.

## Uptake and oxidation of radioactively labeled fatty acids

Uptake and oxidation of radioactively labeled fatty acids was assessed in HUH7 cells pre-incubated with DMSO or with 500 nM TVB 2640 for 24 h. Briefly, radioactive fatty acids and the unlabeled equivalent were conjugated to 0.5 mM FA-free Albumin. Cells were fasted for 1 h in DMEM 0.1% FA-free BSA and then incubated with [14]C-DHA (0.037MBq) or [14]C-oleate (0.037MBq) and 100 μM of the respective unlabeled fatty acid in DMEM 0.1% FA-free BSA for 4 h.

To assess fatty acid uptake, cells were washed and harvested after the 4-h incubation period. In the lysed cell pellets, protein content was determined by the method of Lowry and tracer amounts were counted by scintillation counting.

For fatty acid oxidation analyses, cells were coated with NaOH-soaked filter paper. After 4 h of fatty acid incubation, cells were lysed using perchloric acid and filter papers were allowed to sit for another hour, before they were subjected to scintillation counting.

## Retention/secretion of radioactively labeled fatty acids

Huh7 cells were loaded overnight with a mixture of [3]H-glycerol (0.37MBq) and [14]C-labeled DHA (0.0185MBq) or [3]H-glycerol (0.37MBq) and [14]C-oleate (0.0185MBq) conjugated to 0.5 mM albumin in DMEM 10%FCS. As indicated, cells were either incubated with 500 nM TVB-2640 during lipid loading or not. Loading solution was removed, cells were washed twice with DMEM 1% FCS and then incubated with DMEM 1%FCS, 20U Heparin in the presence or absence of 500 nM TVB2640. After 5 h of secretion, supernatants were collected, and cells were harvested in PBS for the determination of protein concentration and scintillation counting. From both, the cell suspensions and the supernatants, lipids were extracted using the method of Dole[59] as follows: 667 μl of supernatants or cell suspension were mixed with 3333 μl of 80/20 2-propanol/heptane (v/v), with 0.1% H2SO4. Samples were vigorously vortexed and allowed to sit for 10 min. After the addition of 1333 μl heptane and 2 ml water, samples were vortexed and centrifuged (10 min, 1000 g), and the upper organic phase was subjected to scintillation counting.

## Western blot

Cells were cultured as specified, washed with PBS and scraped off in ice-cold cell lysis buffer (50 mM Tris−HCl, pH 8.0; 150 mM NaCl; 1% Nonidet P-40; 0,5% Na-Deoxycholat; 5 mM EDTA, 0,1% SDS), organs were harvested and homogenized in RIPA-buffer supplemented with complete Mini Protease Inhibitors and PhosStop (Roche). Cell lysates or organ lysates were clarified by centrifugation (18,500 g, 10 min, 4 °C) and supernatants were supplemented with sample buffer. Proteins were separated on SDS-polyacrylamide gels and transferred to PVDF membranes using the Transblot Turbo Transfer System (Bio-Rad laboratories). Following blocking (20 mM Tris−HCl, pH 7.4; 150 mM NaCl; 0.1% Tween-20; 5% non-fat dry milk) and washing (20 mM Tris−HCl, pH 7.4; 150 mM NaCl; 0.1% Tween-20), membranes were incubated in primary antibody solution (20 mM Tris−HCl, pH 7.4; 150 mM NaCl; 0.1% Tween-20; 5% BSA or 5% non-fat dry milk) containing the appropriate antibodies (Acetyl-Lys (Cell Signaling; 9441 S; 1:500 dilution), AKT (Cell Signaling; 9272 S; 1:1000 dilution), Calreticulin: (Cell Signaling; #4850; 1:1000.dilution), CHOP (Cell Signaling; 2895; 1:1000 dilution), COXIV (Cell Signaling; #4850; 1:1000 dilution),

e cadherin (Cell Signaling; #3195; 1:1000 dilution), FASN (BD Biosciences; 610962; 1:1000 dilution), Gapdh (Santa Cruz; Sc32233; 1:15000 dilution), γ-tubulin (abcam; Ab179503; 1:2000 dilution), HA (Roche; 12013819001; 1:10000 dilution), Histon h3 (abcam; Ab4729; 1:1000 dilution), P-Ser (Sigma-Aldrich; AB1603; 1:1000 dilution), Rcas (Cell Signaling; #12290; 1:1000 dilution), LDHA (Cell Signaling; 2012; 1:1000 dilution), Malonyl-Lys (Cell Signaling; #14942; 1:1000 dilution), c-MYC (Sigma Aldrich; M5546; 1:100 dilution), Srebp (Invitrogen; MA5-11685; 1:1000 dilution), Ubiquitin (Merck; MAB1510-I; clone Ubi-1; 1:1000 dilution) Membranes were washed and incubated with donkey anti-rabbit IgG Horseradish Peroxidase secondary antibody (GE Healthcare; no. NA934V; 1:7,500 dilution) or with sheep anti-mouse IgG Horseradish Peroxidase secondary antibody (GE Healthcare; no. NA931V; 1:7,500 dilution). After final washing, proteins were visualized using the ChemiDoc MP Imaging System (Bio-Rad laboratories).

## Modeling of a mutant FASN dimer

The hyperacetylated sites identified by our studies were mapped on the surface of the available 3.22 Å mammalian full-length FASN structure, PDB 2VZ8[60]. FASN lysines displaying hyperacetylation are shown as red spheres in one of the chain (residues corresponding to UNP FAS_HUMAN). ACP, acyl carrier protein is shown as a cartoon in the figure. In porcine FAS (UNP A5YV76_PIG), Lys776 is a glutamate, and Lys1927 is an arginine. In the structure, the location of Lys673, Lys776, Lys1582, Lys1704, Lys1927 in the mammalian FASN is shown. The location of hyperacetylated Lys2206, Lys2436, and Lys2449 could not be indicated, because of the unavailability of the thioesterase domain (TE), in the PDB 2VZ8 structure. The figure was prepared using ChimeraX[61].

## Exome sequencing and variant validation

Genomic DNA was extracted from blood samples and trio exome sequencing was performed with DNA samples of the index patient and both healthy parents. For this purpose, coding DNA fragments were enriched with a SureSelect Human All Exon 50 Mb V5 Kit (Agilent) and libraries were sequenced on a HiSeq2500 platform (Illumina). Reads were aligned to the human reference genome (UCSC GRCh37/hg19) using the Burrows-Wheeler Aligner (BWA, v.0.5.87.5). Detection of genetic variation was performed with SAMtools (v.0.1.18), PINDEL (v. 0.2.4t), and ExomeDepth (v.1.0.0). The impact of predicted amino acid substitutions for protein function was assessed by the pathogenicity prediction tools CADD, M-CAP and ClinPred. The *FASN* variant was validated by Sanger-sequencing. Primer pairs were designed to amplify selected coding exons of the candidate gene. Amplicons were directly sequenced using the ABI BigDye Terminator Sequencing kit (Applied Biosystems) and a capillary sequencer (ABI 3500, Applied Biosystems). Sequence electropherograms were analyzed using the Sequence Pilot software (JSI Medical Systems).

## Acetyl-Proteomics

Identification of the proteins captured by the IP was performed by the following steps: The protein band of the SDS-PAGE of the IP eluate fraction was cut and proteins in the band digested with trypsin according to Shevchenko et al.[62]. Briefly, the gel was shrinked and swelled with 100% acetonitrile (MeCN) and 100 mm $NH_4HCO_3$, respectively. Proteins in the gel were reduced with 10 mm dithiothreitol, dissolved in 100 mM NH4HCO3 and alkylated with 55 mM iodacetamide (dissolved in 100 mm NH4HCO3). Tryptic digestion was performed with 8 ng/μL sequencing-grade trypsin, dissolved in 50 mM NH4HCO3 and 10% MeCN at 37 °C for 12 h. Peptide extraction was done with 5% formic acid and 50% MeCN in water and thereafter the peptides were dried. Next, the peptides were dissolved in 0.1% formic acid (20 μL) and injected with a flow rate of 5 μL/min into a nano-LC system (Dionex UltiMate 3000 RSLCnano, Thermo Scientific) containing a trapping column (Acclaim PepMap μ-precolumn, C18, 300 μm

× 5 mm, 5 μm particle size, 100 Å pore size, Thermo Scientific. Buffer A: 0.1% formic acid in H2O; buffer B: 0.1% formic acid in MeCN) coupled to an electrospray ionization (ESI) source, part of a tribrid mass spectrometer equipped with a quadrupole, a linear ion-trap, and an orbitrap (Orbitrap Fusion, Thermo Scientific). Salts and other hydrophilic compounds were washed from the trapping column with 2% buffer 5 min B using a flow rate of 5 μL/min. The desalted tryptic peptides were fractionated with a reversed-phase capillary column (Acclaim PepMap 100, C18, 75 μm × 250 mm, 2 μm particle size, 100 Å pore size, Thermo Scientific). The ESI spray was formed by a fused-silica emitter (I.D. :10 μm, New Objective, Woburn, USA) using a capillary voltage of 1650 V. The positive ion mode was used. The mass spectrometer was operated in the data-dependent acquisition (DDA) / top speed mode. Further parameters were 28% HCD collision energy, an intensity threshold of 2 ×105, and an isolation width of $m/z = 1.6$. For the MS scan a $m/z$ 400–1500 range was chosen, performed every second, with the resolution of 120000 full width at half maximum height (FWHM) at $m/z$ 200, a transient length of 256 ms, a maximum injection time of 50 ms and an AGC target of 2 × 105. Fragment spectra were measured in the ion-trap with a scan rate: 66 kDa/s, a maximum injection time of 200 ms and a AGC target of 1 × 104. With Proteome Discoverer 2.0 (Thermo Scientific) LC-MSMS data were processed. Proteins were identified using the search engine Sequest HT and the human Swiss-Prot protein database (www.uniprot.org). As parameters for the searches a precursor mass tolerance of 10 ppm, a fragment mass tolerance of 0.2 Da, and tryptic/semitryptic digestion were chosen. Two missed cleavages were allowed. As a fixed modification carbamidomethylation of cysteine residues, as a variable modification oxidation of methionine residues and in addition acetylation were applied for the search. A false discovery rate of 1% by using Percolator was applied.

## LC-MS/MS-based analysis of the ubiquitinylation

Directly prior to measurement, lyophilized peptides were resuspended in 0.1% FA to a final concentration of 1 mg/mL. 1 μg of protein was injected into a Dionex Ultimate 3000 UPLC system. For online desalting and purification, a peptide trap (Acclaim PepMap 100, 100 μm × 2 cm, 100 Angstrom pore size, 5 μm particle size (Thermo Fisher Scientific, Bremen, Germany)) was installed in front of a 25 cm C18 reversed-phase analytical column (Acclaim PepMap 100, 75 μm × 50 cm, 100 Angstrom pore size, 2 μm particle size (Thermo Fisher Scientific, Bremen, Germany)). Elution of the peptides occurred with an 80 min gradient with a linearly increasing concentration of buffer B from 2% to 30% in 60 min, rising to 90% for 5 min with equilibration for 10 min at 2% buffer B. Eluted peptides were ionized and desorbed via electrospray ionization, using a spray voltage of 1.8 kV, transferred into a Quadrupole orbitrap hybrid mass spectrometer (QExactive, Thermo Fisher Scientific) and analyzed in data-dependent acquisition (DDA) mode. For each MS1 scan, ions were accumulated for a maximum of 120 milliseconds or until a charge density of 2 × 105 ions (AGC Target) was reached. Fourier-transformation-based mass analysis of the data from the orbitrap mass analyzer was performed covering a mass range of 400–1200 $m/z$ with a resolution of 120,000 at $m/z = 200$. The 15 Peptides most abundant peptides for each precursor scan (Top N mode) with charge states between 2 + – 5+, above an intensity threshold of 1000 were isolated within a 1.6 $m/z$ isolation window and fragmented with a normalized collision energy of 25% using higher energy collisional dissociation (HCD). MS2 scanning was performed, using an orbitrap mass analyzer, at an orbitrap resolution of 15,000 at $m/z = 200$, covering a mass range of 350–1500 $m/z$. Ions were accumulated for 60 ms or to an AGC target of 1 × 105. Already fragmented peptides were excluded for 15 s (Dynamic exclusion).

LC-MS/MS raw data were searched with the Sequest algorithm, integrated into the Proteome Discoverer software, ((v 3.0.0.757), Thermo Fisher Scientific)[63] against a reviewed human Swissprot

database, obtained in December 2021, containing 20365 entries. Carbamidomethylation was set as a fixed modification for cysteine residues. The oxidation of methionine, pyro-glutamate formation at glutamine residues as well as acetylation of the protein N-terminus and Lysine residues were allowed as variable modifications. For the targeted search of Ubiquitinylated Peptides, two Glycine residues, remaining at ubiquitinylated Lysine residues after tryptic digestion, were included as variable modifications. A maximum number of 2 missing tryptic cleavages was set. Peptides between 6 and 144 amino acids where considered. A strict cutoff (FDR < 0.01) was set for Peptide and protein identification. Quantification was performed using the Minora Algorithm, implemented in Proteome Discoverer. Obtained peptide abundances where log2 transformed. Median normalization was performed for each sample.

## Statistical methods

Data were transformed to natural logarithm when necessary to achieve normal distribution and/or homoscedasticity. Statistical analyses using Student's t-test, one-way or two-way ANOVA were performed depending on each experiment and are stated in every figure legend. GraphPad Prism 9 was used for all statistical analyses except states otherwise. For ubiquitinylation proteomic analysis, statistical testing was carried out using the Perseus software[64] (Max Plank Institute for Biochemistry, Version 1.5.8.5.)

## Reporting summary

Further information on research design is available in the Nature Portfolio Reporting Summary linked to this article.

## Data availability

The mass spectrometric proteomics data have been deposited to the ProteomeXchange Consortium via the PRIDE partner repository with the dataset identifier PXD045908. De-identified participant demographics, lipidomic data and the study protocol for the FASCINATE-1 trial will be shared for scientific use, under a material transfer agreement, after review and acceptance of the request by Sagimet Biosciences Inc, with access of up to three years via bd@sagimet.com. All other relevant data of this study are available within the article and its supplementary information. Source data are provided with this paper.

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

## Acknowledgements

The authors are thankful to Katrin Rading, Laura Ehlen, Verena Rickassel, Meike Kröger, Eva-Marie Azizi, Sabine Schröder, Christina Schmidt and Sandra Ehret for excellent technical support. Wild type and *Dgat2*^(-/-) MEFs were kindly provided by Prof. T. Walther and Prof. R.V. Farese Jr. C.S. was supported by the Werner-Otto-foundation, by the DFG (SCHL2276/2-1), the Mühlbauer foundation and by the University Medical Center Hamburg Eppendorf Medical Faculty (TDM-21/06). A.W. is supported by the University Medical Center Hamburg Eppendorf Medical Faculty (TDM-21/06; NWF-20/07) and the Mühlbauer-foundation and the DFG (335447717 - SFB 1328). F.H. was supported by the Werner-Otto-foundation. S.A. and C.F.S. are supported by NIH DK20579. A.W.F. received support from the German Research Council (DFG) (FI 2476/1-1) and the Human Frontier Science Program (HFSP). J.H., C.S., A.W., and L.S. are supported by a grant funded by the DFG (450149205-TRR333/1). In addition, A.W., J.H., and L.S. H.Sc. are supported by the State of Hamburg (LFF-FV75). This study was supported by grants from the Deutsche Forschungsgemeinschaft (DFG) (INST 337/15-1, INST 337/16-1, INST 152/837-1 and INST 152/947-1 FUGG). Graphical Models were created with BioRender.com. We acknowledge financial support from the Open Access Publication Fund of UKE - Universitätsklinikum Hamburg-Eppendorf and DFG – German Research Foundation.

## Author contributions

A.W. generated data, analyzed data, and wrote the manuscript. J.R., I.E., S.Y.L.P, M.M., H.V., A.W.F., S.A., K.K., M.S., U.K., L.S., H.Sc., J.S., T.B., M.Heine, M.P., F.H., S.K., M.M.F., A.N., P.P., H.Si., K.T., M.Hempel, C.F.S., and J.H., generated or analyzed data and edited the manuscript. M.O. analyzed and supervised biomarker components of the FASCINATE-1

trial and edited the manuscript. C.K. analyzed data and wrote the manuscript. C.S. initiated and designed studies, analyzed data, directed the overall execution of studies and wrote the manuscript. C.S is the guarantor of this work and, as such, has full access to all the data of the study and takes responsibility for the integrity of the data and the accuracy of the data analysis.

## Funding

## Competing interests
M.O. is an employee of Sagimet. C.S. has been an invited speaker for Daiichi Sankyo. All other authors declared no conflicting interests.
