## [Peer Review File · Nature Communications]

Fatty acid synthesis suppresses dietary polyunsaturated fatty acid useEditorial Note: This manuscript has been previously reviewed at another journal that is not operating a transparent peer review scheme. This document only contains reviewer comments and rebuttal letters for versions considered at *Nature Communications*.

REVIEWER COMMENTS

Reviewer #1 (Remarks to the Author):

The authors have addressed my main concerns.

Reviewer #2 (Remarks to the Author):

We appreciate the authors' efforts in addressing our comments. However, we feel that our main concern regarding the specificity of their mechanism has not been adequately addressed. To establish the specificity of PUFA uptake and partitioning into TAGs compared to SFAs or MUFAs, we suggest conducting stable isotope tracing experiments to demonstrate the effect of DNL inhibition. This would help to prove that DNL inhibition specifically results in the uptake and partitioning of PUFA into TAGs.

Regarding Comment 4, we thank the authors for acknowledging that HDAC3 targeting is not a viable option for restoring FASN function. However, it would be valuable if they could demonstrate the improvement in FASN protein stability through HDAC3 overexpression using CHX, as shown in Fig. 1b.

We agree with the authors on the relative increase in PUFA levels resulting from DNL inhibition (Comments 5-7 and 9-15). However, we still have concerns regarding the specificity of this mechanism. It is important for the authors to address our point 8 to establish the specificity of their mechanism.

In Comment 8, we express our reservations about the use of radiation, as it prevents a direct comparison between MUFA (or SFAs) and PUFAs in the context of DNL inhibition. We suggest that the authors employ an equimolar treatment of stably labeled SFAs, MUFAs, and PUFAs to demonstrate two key points: 1) preferential uptake of PUFAs and 2) preferential channeling of PUFAs into TAGs. Additionally, the authors should consider controlling for FAO, as different FAs have varying susceptibilities to oxidation.

Lastly, we appreciate the authors' correction of the issue mentioned in Comment 16.

Reviewer #3 (Remarks to the Author):

The reviewer thanks the authors for their responses

Although there are still areas of uncertainty, I think there is enough interesting material in here for this to now appear without further alterations

Reviewer #4 (Remarks to the Author):

The authors have adequately addressed the reviewer's concern and I did not have additional questions.

Reviewer #1 (Remarks to the Author):

The authors have addressed my main concerns.

Response: We thank Reviewer #1 for the discussion and additions to this manuscript.

Reviewer #2 (Remarks to the Author):

We appreciate the authors' efforts in addressing our comments.

Response: We thank the reviewer for the valuable suggestions.

However, we feel that our main concern regarding the specificity of their mechanism has not been adequately addressed. To establish the specificity of PUFA uptake and partitioning into TAGs compared to SFAs or MUFAs, we suggest conducting stable isotope tracing experiments to demonstrate the effect of DNL inhibition. This would help to prove that DNL inhibition specifically results in the uptake and partitioning of PUFA into TAGs.

Response: We thank the reviewer for the suggestions. We now added the suggested data to Fig.3d and Fig.S4f (see also comment below).

Regarding Comment 4, we thank the authors for acknowledging that HDAC3 targeting is not a viable option for restoring FASN function. However, it would be valuable if they could demonstrate the improvement in FASN protein stability through HDAC3 overexpression using CHX, as shown in Fig. 1b.

Response: We thank the reviewer for the suggestion. We added the requested data showing unchanged FASN abundance of FASN-R2177C in 0h vs 6h CHX during co-expression of HDAC3 to Supplemental Figure 1j indicating that HDAC3 co-expression might stabilize FASN during CHX treatment.

We agree with the authors on the relative increase in PUFA levels resulting from DNL inhibition (Comments 5-7 and 9-15). However, we still have concerns regarding the specificity of this mechanism. It is important for the authors to address our point 8 to establish the specificity of their mechanism. In Comment 8, we express our reservations about the use of radiation, as it prevents a direct comparison between MUFA (or SFAs) and PUFAs in the context of DNL inhibition. We suggest that the authors employ an equimolar treatment of stably labeled SFAs, MUFAs, and PUFAs to demonstrate two key points: 1) preferential uptake of PUFAs and 2) preferential channeling of PUFAs into TAGs. Additionally, the authors should consider controlling for FAO, as different FAs have varying susceptibilities to oxidation.

Response: We thank the reviewers to further explain their reservations. We performed the suggested experiments using stable isotope tracing. In line with the *in vitro* and *in vivo* radiation data, stable isotope tracing in HUH7 cells shows higher incorporation of DHA, but not oleate or palmitate, into secreted TAG (new Fig.3d) after FASN inhibition, confirming our previous findings. In cell pellets, FASN inhibition even suppressed exogenous palmitate incorporation and resulted in higher DHA and oleate incorporation suggesting a higher DHA flux through the TAG compartment. When we controlled for fatty acid oxidation, as suggested by the reviewers, DHA incorporation was higher than oleate in both cell pellets and supernatant after TVB-2640 treatment (etomoxir+TVB-2640 DHA vs oleate and/or palmitate) (new Fig.3d/Fig.S4f).

This further emphasizes the impact of FASN on the PUFA TAG incorporation and explaining the higher compositional abundance of DHA after FASN inhibition. We added these additional mechanistical insights to the manuscript and thank the reviewers for their valuable suggestions.

Lastly, we appreciate the authors' correction of the issue mentioned in Comment 16.

Response: We thank the reviewer for the suggestions, criticism, comments which very much helped the manuscript to improve. We appreciate the efforts of the reviewer to deeply understand the nature of the manuscript.

Reviewer #3 (Remarks to the Author):

The reviewer thanks the authors for their responses

Although there are still areas of uncertainty, I think there is enough interesting material in here for this to now appear without further alterations

Response: We thank Reviewer #3 for the fruitful discussion and additions to this manuscript.

Reviewer #4 (Remarks to the Author):

The authors have adequately addressed the reviewer's concern and I did not have additional questions.

Response: We thank Reviewer #4 for the efforts to improve the quality of the manuscript.

REVIEWERS' COMMENTS

Reviewer #2 (Remarks to the Author):

We thanks the authors for addressing our commments.